# Cardiac interventions in Wales: A comparison of benefits between NHS Wales specialties

**Gareth Davies**[1]*, **Ashley Akbari**[1], **Rowena Bailey**[1], **Lloyd Evans**[2], **Kendal Smith**[3], **Jonathan Goodfellow**[2], **Michael Thomas**[4], **Kerryn Lutchman Singh**[3]

**1** Population Data Science, Swansea University Medical School, Faculty of Medicine, Health & Life Science, Swansea University, Swansea, Wales, **2** NHS Wales Executive, Wales Cardiovascular Network, Cardiff, Wales, **3** Welsh Health Specialised Services Committee, Pontypridd, Wales, **4** Hywel Dda University Health Board, Hafan Derwen, Carmarthen, Wales

* g.i.davies@swansea.ac.uk

## Abstract

### Objectives

The study aimed to assess if specialised healthcare service interventions in Wales benefit the population equitably in work commissioned by the Welsh Health Specialised Services Committee (WHSSC).

### Approach

The study utilised anonymised individual-level, population-scale, routinely collected electronic health record (EHR) data held in the Secure Anonymised Information Linkage (SAIL) Databank to identify patients resident in Wales receiving specialist cardiac interventions. Measurement was undertaken of associated patient outcomes 2-years before and after the intervention (minus a 6-month clearance period on either side) by measuring events in primary care, hospital attendance, outpatient and emergency department. The analysis controlled for comorbidity (Charlson) and deprivation (Welsh Index of Multiple Deprivation), stratified by admission type (elective or emergency) and membership of top 5% post-intervention costs. Costs were estimated by multiplying events by mean person cost estimates.

### Results

We identified 5,999 percutaneous coronary interventions (PCI) and 1,640 coronary artery bypass graft (CABG) between 2014-06-01 to 2020-02-29. The ratio of emergency to elective interventions was 2.85 for PCI and 1.04 for CABG. In multivariate analysis significant associations were identified for comorbidity (OR = 1.52, CI = (1.01–2.27)), deprivation (OR = 1.34, CI = (1.03–1.76)) and rurality (OR = 0.81, CI = (0.70–0.95)) for PCI interventions, and comorbidity (OR = 1.47, CI = (1.10–1.98)) for CABG. Higher costs post-intervention were associated with increased comorbidity for PCI and CABG in the top 5% cost groups, but for PCI this was not seen outside the top 5%. For PCI, moderate cost increase was associated with increased deprivation, but the picture was more mixed following CABG interventions. For both interventions, lower costs post intervention were seen in rural locations.

**Data Availability Statement:** The data used in this study are available in the SAIL Databank at Swansea University, Swansea, UK, but as restrictions apply, they are not publicly available. All

proposals to use SAIL data are subject to review by an independent Information Governance Review Panel (IGRP). Before any data can be accessed, approval must be given by the IGRP. The IGRP carefully considers each project to ensure the proper and appropriate use of SAIL data. When access has been granted, it is gained through a privacy-protecting trusted research environment (TRE) and remote access system referred to as the SAIL Gateway. SAIL has established an application process to be followed by anyone who would like to access data via SAIL at https://www.saildatabank.com/application-process.

**Funding:** The work was funded by the Welsh Health Specialised Services Committee (WHSSC). The funders had no role in study design, data collection and analysis, decision to publish, or preparation of the manuscript.

**Competing interests:** The authors have declared that no competing interests exist.

## Conclusion

We identified and compared health outcomes for selected specialist cardiac interventions amongst patients resident in Wales, with these methods and analyses, providing a template for comparing other cardiac interventions.

## Introduction

This study was commissioned by the Welsh Health Specialised Services Committee (WHSSC) in December 2020. The WHSSC includes representatives from all health boards in Wales and has the purpose of ensuring health care is delivered equally to the population of Wales [1]. Healthcare focus has traditionally been placed on annual incremental increases in funding, meaning equity of access between regions for the same service has not been routinely addressed [2]. Some variation was therefore anticipated. It was desired to see whether, in the patient pathways for Welsh patients, there was a significant variation in access rates between differing health interventions and conditions and against the expected background level, which may indicate inequity of access.

Cost analyses in general for healthcare provisions are not as widespread as desired [3, 4]. Cost profiling within particular areas would inform cost-effectiveness, as high-cost models may deliver better value than low-cost models in certain areas of health [5]. Area clusters would also be of interest, where geographical and population density may influence ease of access to health services, as seen in other studies [6, 7]. Varying disease burden and reporting of disease may factor amongst different deprivation levels [8], and patients with higher comorbidity are likely to feature in excess events. Predicting the patient pathway model over time is likely to provide better patient outcomes [9, 10] and is becoming more of a focus [11].

This study aimed to assess if specialised healthcare service interventions in Wales benefit the population equally, by comparing costs of healthcare between differing demographic and socio-economic groups. Health resource usage was compared pre- and post-intervention to understand what impact each treatment had on local health service use. The ability to add cost information enabled where on the patient pathway most benefit could be gained in terms of change or investment.

## Method

This study developed a method to evaluate medium and long-term benefits in a range of specialities, monitoring changes in resource use over time, comparing outcomes from alternative interventions, and measuring pressure on secondary services. To compare the effect of interventions, primary and secondary healthcare events were measured on either side of the first intervention date. Associated costs were calculated, and the pathway type was categorised into either elective or emergency for hospital admissions.

### Data sources

This study utilised the Secure Anonymised Information Linkage (SAIL) Databank in Swansea, a trusted research environment (TRE) providing linked individual-level, anonymised population-scale data on the population of Wales, UK. The SAIL Databank contains a collection of anonymised linked data sources, including routinely collected health and socioeconomic data at an individual level, encrypted by SAIL's trusted third party, Digital Health and Care Wales (DHCW) [12–16].

**Table 1. List of healthcare interventions.**

| Healthcare intervention |
| --- |
| Percutaneous coronary intervention [PCI] |
| Transcatheter Aortic Valve Implantation [TAVI] |
| Electrophysiology [EP] ablations—standard |
| Electrophysiology [EP] ablations—complex |
| Electrophysiology [EP] study |
| Cardiac device implants (pacemaker or defibrillator) |
| Cardiac surgery—coronary artery bypass graft [CABG] |
| Cardiac surgery—valve replacement |

The following SAIL data sources were available to the project following approval from the SAIL independent Information Governance Review Panel (IGRP):

- Annual District Death Extract (ADDE).

- Emergency Department Data Set (EDDS).

- Outpatient Database for Wales (OPDW).

- Patient Episode Database for Wales (PEDW).

- Welsh Cancer Intelligence & Surveillance Unit (WCISU).

- Welsh Demographic Service Dataset (WDSD).

- Welsh Longitudinal General Practice (WLGP).

- Welsh Results Reporting Service (WRRS).

The interventions examined are listed in Table 1.

Interventions and conditions were identified (see supplementary material S3 and S4 Tables) using Healthcare Resource Group (HRG) [17], Operating Procedure Codes Supplement (OPCS-4) [18], Read and International Classification of Diseases (ICD-10) codes [19]. The WLGP data were used to identify interactions with primary care using Read codes [20]. The Read codes were selected from pre-defined Quality Outcome Framework (QOF) code lists [21]. The QOF provided a financial incentive for GPs to record data for conditions listed on the QOF, therefore more likely to provide a good level of coverage. QOF has recently (post-2019) been superseded by the Quality Assurance and Improvement Framework (QAIF) [22]. Patient events for the interventions were filtered to remove the following conditions for each intervention (see Table 2 below).

**Table 2. Conditions filtered from intervention events.**

| Intervention | Filtered condition |
| --- | --- |
| PCI | CHD |
| TAVI | Other circulation problems |
| EP ablations—standard | Problems of rhythm |
| EP ablations—complex | Problems of rhythm |
| EP study | Problems of rhythm |
| Cardiac device | CHD+Problems of rhythm+ Other circulation problems |
| Cardiac surgery—CABG | CHD+Other circulation problems |
| Cardiac surgery—Valve | Other circulation problems |

**Table 3. ALF status code in SAIL.**

| Field name | Field description | Field value | Field value description |
|---|---|---|---|
| ALF_STS_CD | Anonymised linkage field status code | 1 | NHS Number passes check digit test |
| | | 4 | Surname, First Name, Post Code, Date of Birth and Sex Code match exactly to AR |
| | | 39 | Surname, Post Code, Date of Birth and Sex Code match exactly to AR, First Name matches on Lexicon (known variants) or Fuzzy Matching probability $>= 0.9$ |

## Ethics approval and consent to participate

Approval for the use of anonymised data in this study, provisioned within the Secure Anonymised Information Linkage (SAIL) Databank, was granted by an independent Information Governance Review Panel (IGRP) under project 1297. The IGRP has a membership comprised of senior representatives from the British Medical Association (BMA), the National Research Ethics Service (NRES), Public Health Wales and Digital Health and Care Wales (DHCW). The usage of additional data was granted by each respective data owner. The SAIL Databank is compliant with General Data Protection Regulations (GDPR) and the UK Data Protection Act.

## Cohort

All SAIL data sources contain a unique anonymised individual identifier, known as the Anonymised Linkage Field (ALF) [12, 13]. The quality of this process is assessed via a linkage certainty percentage, and is reflected in the ALF status field. In extracting the initial cohort, the person identifiers (ALF_PE) were extracted from each data source and filtered to include only those having good linkage status (see Table 3).

The cohort was then further filtered to events which occurred within the study period. This process is seen in Fig 1.

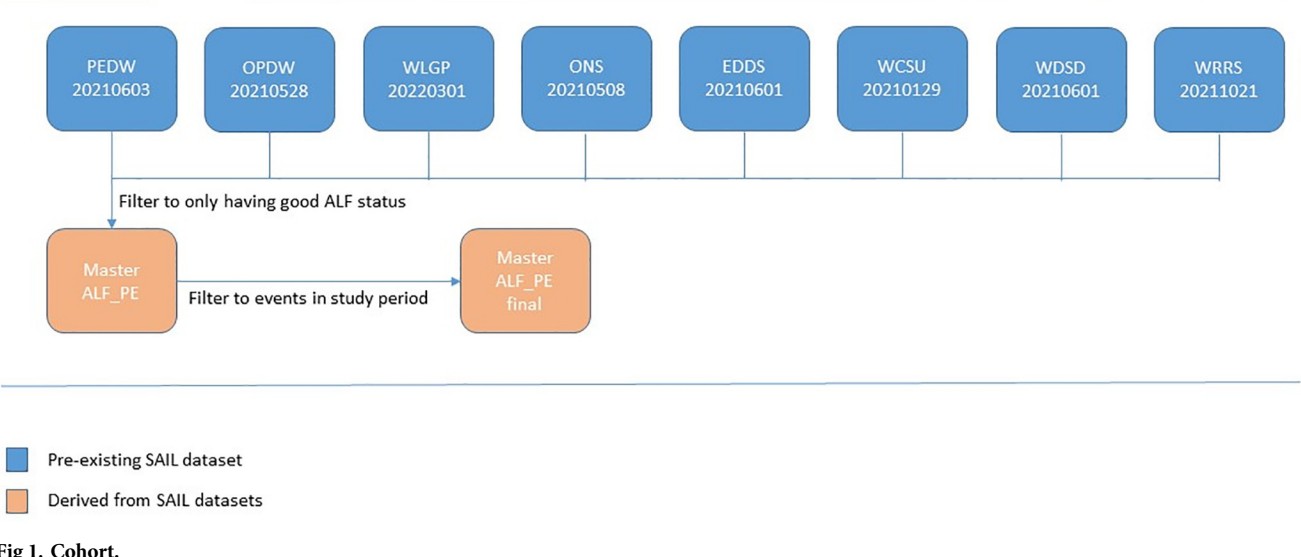

**Fig 1. Cohort.**

## Patient pathway (Study outcomes)

To measure primary and secondary care (emergency department, hospital admissions and outpatient attendances) usage, records from the WLGP, EDDS, PEDW and OPDW data sources were extracted. These were filtered to dates occurring during the study period.

Where secondary diagnoses were present in the PEDW data, only the primary diagnosis was selected. The PEDW data span a period of time consisting of spells and episodes with a start and end date, whereas the WLGP, OPDW and EDDS events have a single event date. A PEDW spell consists of one or more episodes. For this analysis, we used PEDW episodes to increase granularity. PEDW episodes were converted into bed days by subtracting the episode end date from the start date. The admission type in PEDW was determined as elective or emergency using the admission method code. Where admission type could not be determined, these were labelled as unknown. After categorising the interventions as elective or emergency, the ratio of elective to emergency was calculated.

## Covariates

In selecting covariates, we considered how best to measure the patient pathway and variation in healthcare usage. We also chose covariates which are of interest to service commissioners. The study adjusted for age at event, sex, deprivation, rurality of location, comorbidity, type of admission (elective/emergency/unknown), any cost prior to intervention, and outlier status (in the top 5% cost). Geographical location was determined from the Lower-layer Super Output Area (LSOA) version 2011 boundaries [23]. LSOA are statistically generated areas containing approximately 1,500 people, which are larger than (for example) postcodes. There are 1,909 Welsh LSOAs in total. LSOA 2011 was used to determine deprivation levels via the Welsh Index of Multiple Deprivation (WIMD) 2019 quintiles [24] and urban/rural categorisation [25]. Comorbidity was assessed by weighted Charlson comorbidity score [26].

## Mortality

Mortality marks the end of the patient pathway if occurring within two years post-intervention. Mortality was sourced from ADDE and WDSD, with priority given to ADDE in the event of a conflict. Where there was no ADDE date of death, WDSD was used if present. The ADDE tends to have a longer data lag than WDSD, so it is not unusual to have some deaths in WDSD that are not present in ADDE, although the cause of death is only available from ADDE. Date of death was used to derive individual measure of follow-up per person to facilitate comparison of aggregated counts of events between persons within the study.

## Data extraction

ICD-10 codes were used to identify conditions in PEDW. Limited ICD-10 codes are also available in OPDW, so we were able to further supplement the PEDW results with OPDW. The process of creating the data extraction, which was used for the analysis is outlined in Fig 2.

The secondary care data within SAIL is population level coverage for the resident population of Wales, all records relating to interventions and associated follow-up services delivered in Welsh settings are captured in the data, as well as records of interactions between residents of Wales attending English NHS settings. The data does not include records relating to private surgeries or procedures. The data relating to primary care covers approximately 82% of the resident population, as such, there may be records relating to GP visits not available in SAIL. The absence of records for individuals is assumed to be a true non-event, and not considered as missing data. Thus, data imputation methods were not considered.

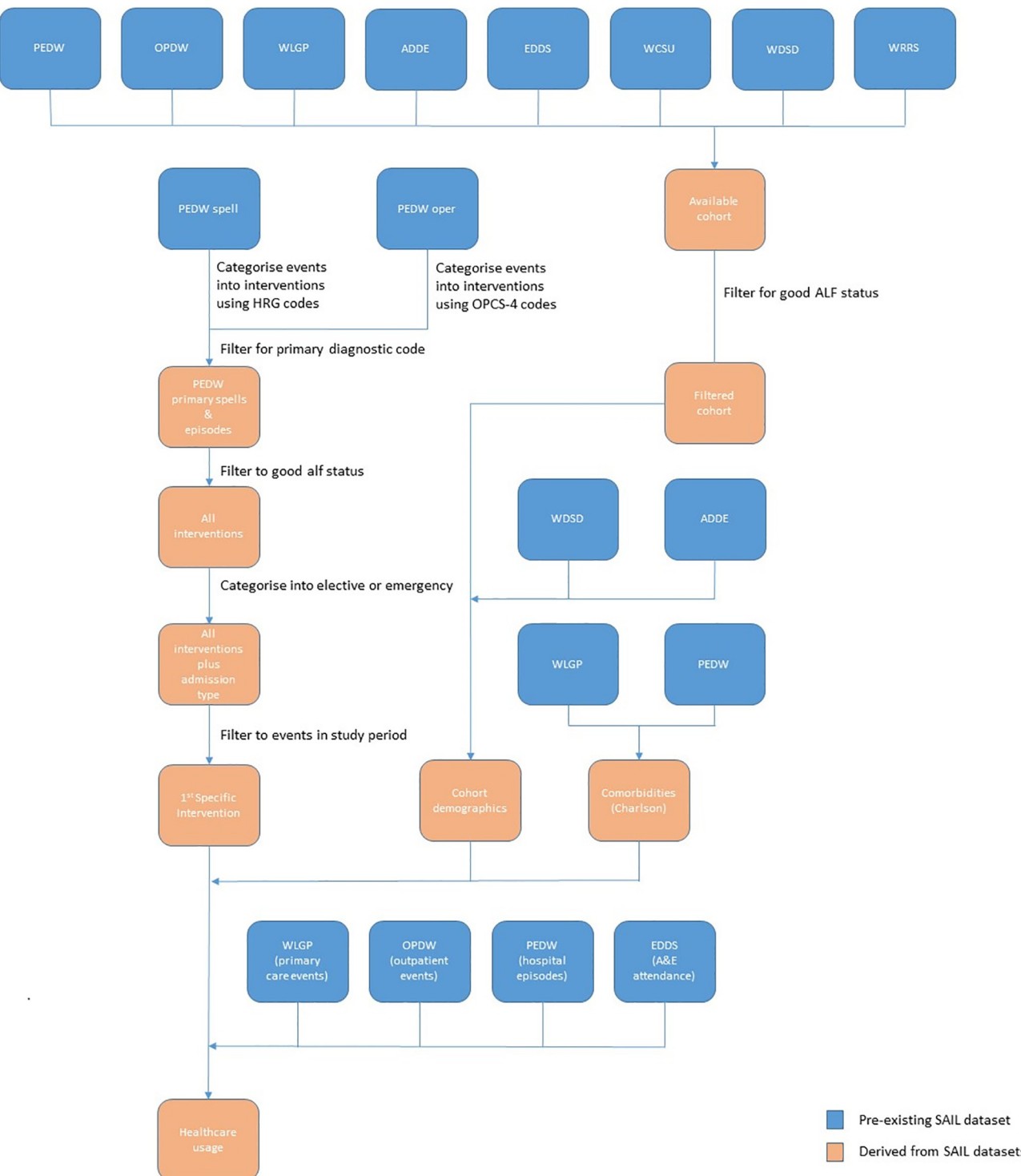

**Fig 2. Data linkage for measurement of interventions health care usage.**

SAIL provides population level coverage for the resident population of Wales, all records relating to interventions and services delivered in Welsh settings are captured, as well as records of interactions between residents of Wales attending English NHS settings. The data does not include records relating to private surgeries or procedures.

## Study period

Data for each intervention were well populated from June 2014 onwards. We curtailed data until the end of February 2020 to avoid the COVID-19 pandemic, after which data were likely to be atypical [27]. Therefore, the study period looked at the complete years 2015 to 2019 inclusive. When measuring pre and post-intervention events, a 'washout' period of 6 months (Fig 3) was applied on either side of the earliest intervention date to provide a clear separation between the two periods being compared, and exclude activity occurring around the intervention period. Therefore to allow 18 months follow-up on either side of an intervention, a study period of June 2016 to February 2018 was applied to the intervention date.

## Cost

The cost for primary care (WLGP), hospital admission (PEDW), outpatient (OPDW) and emergency department (EDDS) was calculated by multiplying the event numbers by the unit cost for each category of provision. HRG codes were used to identify the interventions, but healthcare usage was measured using event numbers in WLGP, PEDW, OPDW and EDDS. The unit costs for PEDW, OPDW and EDDS are derived from WHSSC internal reports, and WLGP cost is derived from Punekar et al. [28]. Unit costs are detailed in Table 4.

## Statistical analysis

Event counts and related costs for the different pathways (elective and emergency hospital bed days, GP interaction, emergency department (ED) attendance, outpatient events) were stratified by sex, age group, social deprivation category (WIMD quintile), number of comorbidities before intervention (24 to 6 month prior), and compared pre and post-intervention. Zero cost

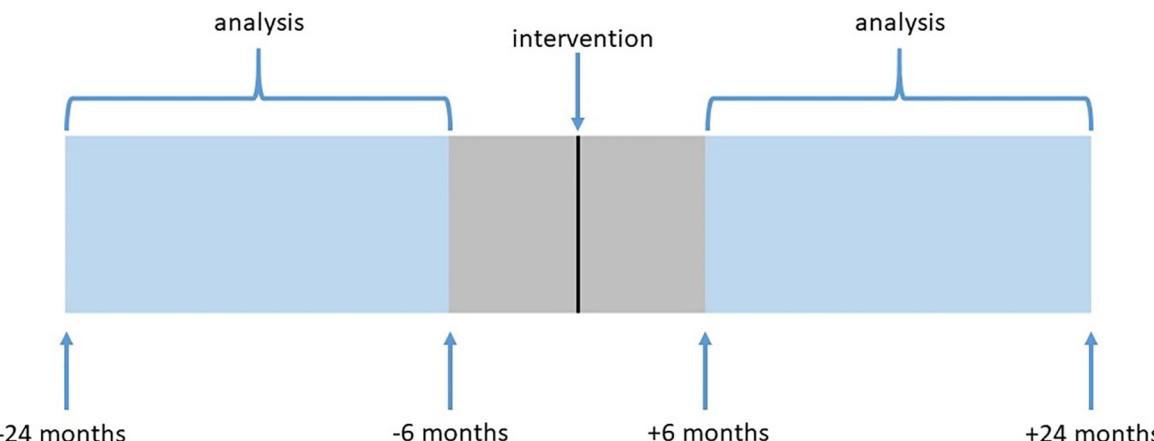

**Fig 3. Interventions washout period.**

**Table 4. Unit cost of NHS healthcare provision.**

| Healthcare setting | Unit cost | Measure |
|---|---|---|
| WLGP event | £36 per event | WLGP events |
| PEDW admission bed days | £398 per day | PEDW episode length |
| OPDW attendance | £143 per event | OPDW events |
| EDDS attendance | £188 per visit | EDDS events |

analysis was also compared to non-zero cost, as many people incurred zero events under certain categories.

To identify factors associated with high and zero cost, univariate and multivariate logistic regression was carried out on the total cost of healthcare usage to identify characteristics of patients with the highest costs (top 5%) compared to; those with zero costs and; everyone else. The person events were categorised into the top 5% costs bracket and by admission type sub-categories (elective/emergency) where numbers were sufficient. Separate models were constructed for each intervention and for the individual cost comparisons. STATA software version 15 was used to run the analyses.

## Results

The total number of people in all data sources (ADDE, EDDS, OPDW, PEDW, OPDW, WCSU, WDSD, WLGP and WRRS, over all time periods, having good linkage (Anonymised linkage field (ALF) status = 1,4 or 39) was 5,933,692.

The number of people identified from each data source is shown in Fig 4.

Of the number of interventions carried out amongst the cohort, PCI was the most represented, with 5,999 procedures identified. The highest number of interventions were found in PCI (= 5,999) and cardiac surgeries (CABG = 1,640, then valve replacement = 918). TAVI interventions were the least numerous at 125 procedures identified. EP complex and EP studies were also low in numbers, meaning regression models were more limited for these groups.

The Elective:Emergency ratios varied from 0.35 to 36.6, with the PCI and CABG procedures manifest proportionally more as emergency interventions, whereas the other interventions were more elective. PCI intervention had nearly three times more emergency than elective. CABG interventions were the only other type to have more emergency than elective. Electrophysiology interventions had the highest elective:emergency ratio (between 6.40 and 36.6 times more elective).

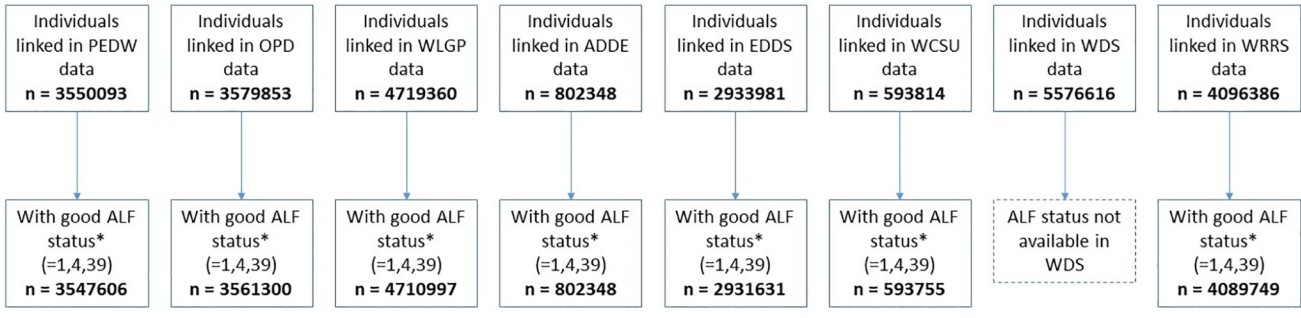

**Fig 4. Number of people in data sources having good linkage.** * ALF (anonymised linkage field) status code indicates quality of matching. Values 1,4,39 indicate good matching.

**Table 5. Number of interventions with associated elective:emergency ratio.**

| Intervention | Number of patients having intervention (June16-Feb18) | Elective:Emergency ratio |
|---|---|---|
| Electrophysiology [EP] ablations—complex | 264 | 36.6 |
| Electrophysiology [EP] ablations—standard | 609 | 14.1 |
| Electrophysiology [EP] study | 150 | 6.40 |
| Cardiac surgery—coronary artery bypass graft [CABG] | 1,640 | 0.96 |
| Cardiac device implants (pacemaker or defibrillator) | 783 | 1.67 |
| Percutaneous coronary intervention [PCI] | 5,999 | 0.35 |
| Transcatheter Aortic Valve Implantation [TAVI] | 125 | 1.51 |
| Cardiac surgery—valve replacement | 918 | 3.30 |

The number of people who received each type of intervention during the period of study (1st June 2016 to 29th February 2018), along with their elective:emergency ratios are shown in Table 5.

The relative costs in each healthcare setting per intervention before and after the 1st intervention are shown in Figs 5–8 and detailed in Tables 6–9. Emergency bed days (PEDW emergency) account for the largest proportion of pre and post-intervention costs for emergency and elective patients. The cost of accident and emergency (EDDS) attendances were higher for electrophysiology interventions in comparison to other intervention types. Primary care (WLGP) was the lowest cost burden, whereas hospital bed days (PEDW) accounted for the overwhelming majority of costs.

Univariate and multivariate logistic regression models are detailed in S1 and S2 Tables of the supplementary material. Figs 9 and 10 display odds ratios with 95% confidence intervals for associated risk factors for the most populous interventions (PCI and CABG). Significant associations to the 5% level are highlighted in bold type.

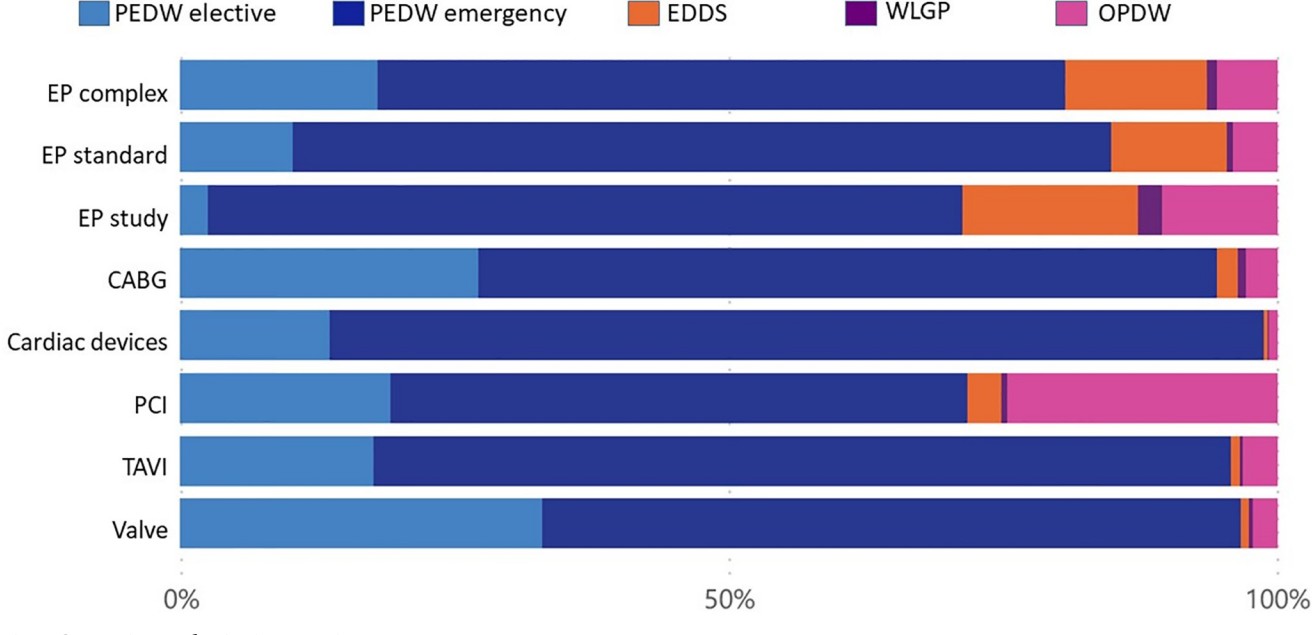

**Fig 5. Cost ratio pre elective intervention.**

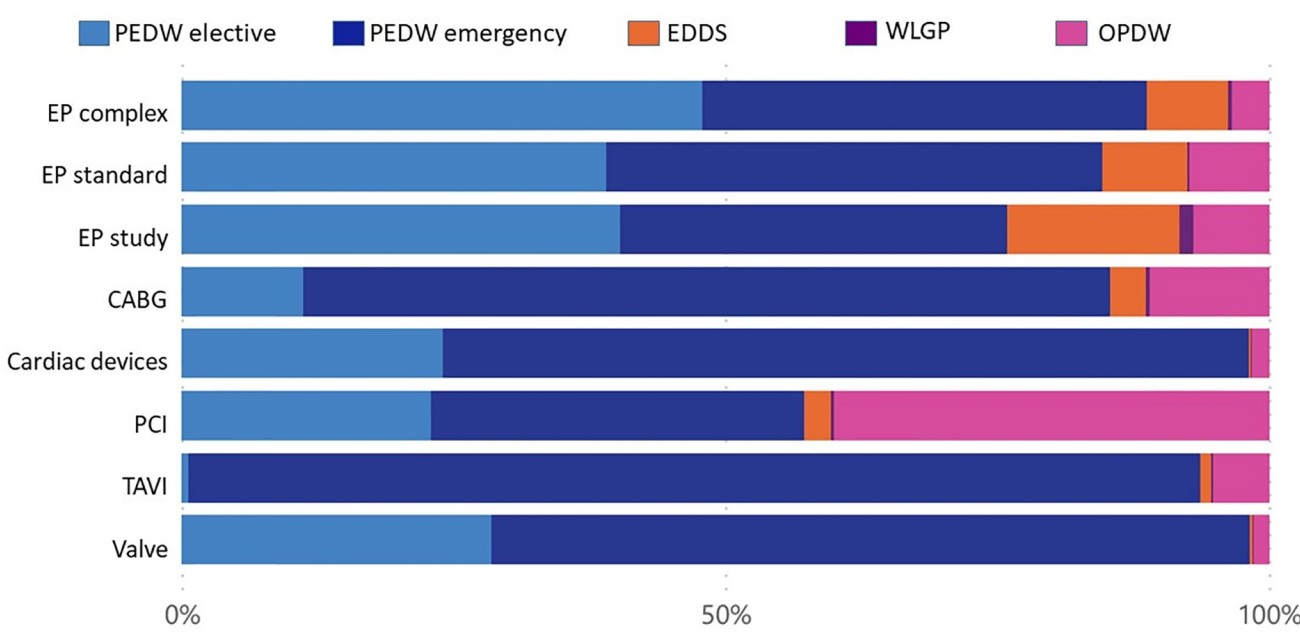

**Fig 6. Cost ratio post elective intervention.**

For PCI intervention, the male/female split showed a slight trend towards more females in the top 5% costs for emergency interventions, but no trend in total cost groups. Age categories revealed a mixed picture. Deprivation showed a slight trend towards most deprived for all top 5% cost categories, and total cost categories. Lower costs were seen in more rural locations. Higher comorbidity was present in patients in the top 5% of total costs group, but reduced in

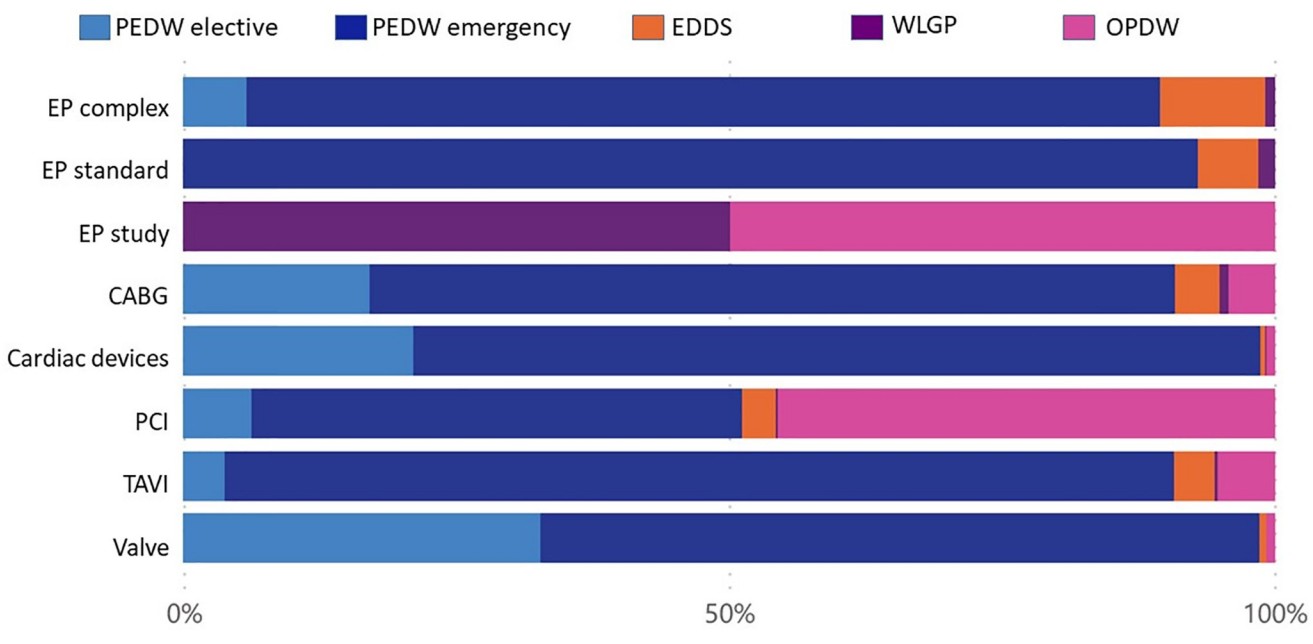

**Fig 7. Cost ratio pre emergency intervention.**

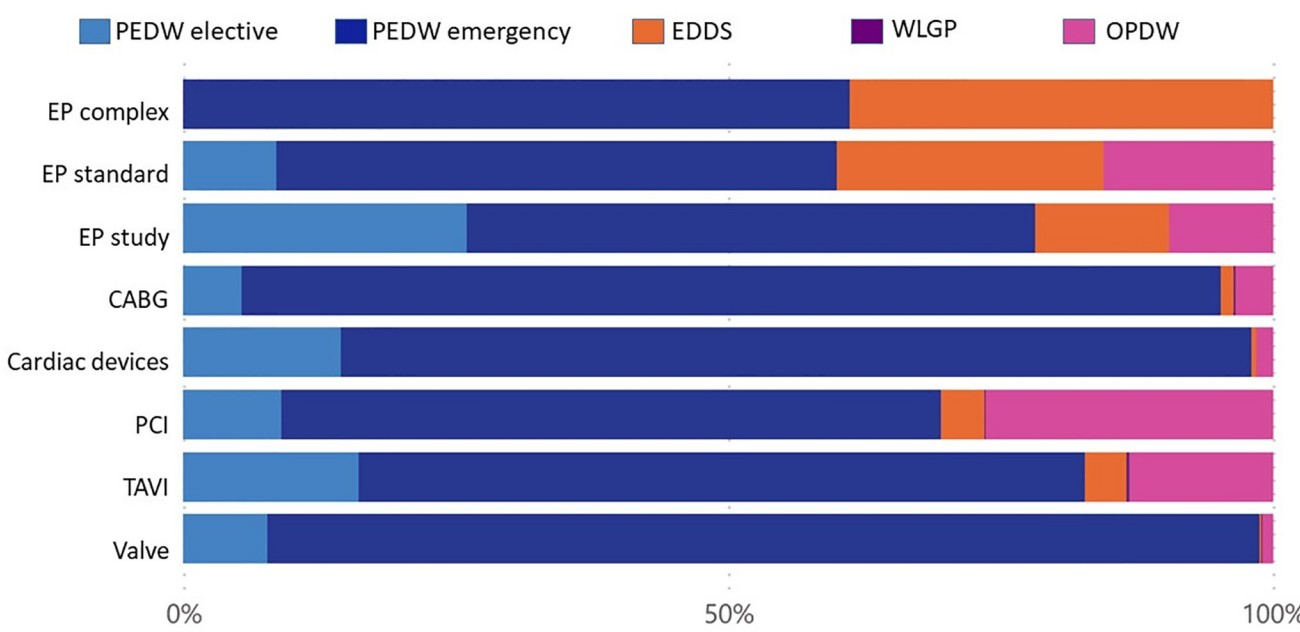

**Fig 8. Cost ratio post emergency intervention.**

**Table 6. Intervention by admission type.**

| Intervention | Admission Type (%) | | |
|---|---|---|---|
| | Elective | Emergency | Uncategorised |
| Electrophysiology [EP] ablations—complex | 96.97 | 2.65 | 0.38 |
| Electrophysiology [EP] ablations—standard | 92.78 | 6.57 | 0.66 |
| Electrophysiology [EP] study | 85.33 | 13.33 | 1.33 |
| Cardiac surgery—coronary artery bypass graft [CABG] | 46.77 | 48.96 | 4.27 |
| Cardiac device implants (pacemaker or defibrillator) | 61.17 | 36.53 | 2.30 |
| Percutaneous coronary intervention [PCI] | 25.50 | 72.65 | 1.85 |
| Transcatheter Aortic Valve Implantation [TAVI] | 59.20 | 39.20 | 1.60 |
| Cardiac surgery—valve replacement | 75.60 | 22.88 | 1.53 |

**Table 7. Intervention by number of comorbidities.**

| Intervention | Number of comorbidities (%) | | | | | |
|---|---|---|---|---|---|---|
| | 0 | 1 | 2 | 3 | 4 | 5 |
| Electrophysiology [EP] ablations—complex | 77.65 | 14.02 | 5.30 | 2.65 | 0.38 | |
| Electrophysiology [EP] ablations—standard | 66.50 | 22.50 | 5.42 | 2.96 | 1.64 | 0.99 |
| Electrophysiology [EP] study | 74.67 | 15.33 | 8.00 | 0.67 | 0.67 | 0.67 |
| Cardiac surgery—coronary artery bypass graft [CABG] | 55.98 | 23.90 | 9.63 | 5.91 | 2.56 | 2.01 |
| Cardiac device implants (pacemaker or defibrillator) | 43.30 | 22.86 | 15.45 | 8.05 | 6.00 | 4.34 |
| Percutaneous coronary intervention [PCI] | 67.88 | 18.42 | 6.70 | 3.82 | 1.50 | 1.68 |
| Transcatheter Aortic Valve Implantation [TAVI] | 40.00 | 20.80 | 15.20 | 12.00 | 5.60 | 6.40 |
| Cardiac surgery—valve replacement | 53.59 | 24.84 | 10.78 | 5.77 | 2.07 | 2.94 |

**Table 8. Intervention by Welsh Index of Mass Deprivation (WIMD) quintile.**

| Intervention | WIMD category (%) | | | | |
|---|---|---|---|---|---|
| | **1. Most deprived** | **2** | **3** | **4** | **5. Least deprived** |
| Electrophysiology [EP] ablations—complex | 7.20 | 14.77 | 23.86 | 27.27 | 26.89 |
| Electrophysiology [EP] ablations—standard | 13.14 | 17.57 | 25.45 | 23.15 | 20.69 |
| Electrophysiology [EP] study | 20.00 | 20.00 | 24.00 | 14.00 | 22.00 |
| Cardiac surgery—coronary artery bypass graft [CABG] | 17.87 | 17.50 | 21.34 | 22.01 | 21.28 |
| Cardiac device implants (pacemaker or defibrillator) | 20.43 | 20.05 | 20.43 | 18.52 | 20.56 |
| Percutaneous coronary intervention [PCI] | 21.10 | 20.37 | 20.67 | 18.65 | 19.20 |
| Transcatheter Aortic Valve Implantation [TAVI] | 15.20 | 23.20 | 20.80 | 18.40 | 22.40 |
| Cardiac surgery—valve replacement | 19.61 | 17.65 | 19.06 | 20.48 | 23.20 |

total cost categories in the adjusted analysis. Admission type showed a mixed picture where numbers were sufficient. Cost prior to intervention was associated with higher cost after intervention. Outliers were also associated with higher cost.

For CABG interventions, fewer females were seen in top 5% groups, in contrast to PCI. Again, age group showed a mixed picture. Deprivation, rurality, comorbidity and prior cost showed similar trends to PCI. Where admission type was emergency, higher cost post intervention was seen. Outliers showed more cost except in emergency interventions for both top 5% and total cost categories.

## Discussion

Our results demonstrate that the most frequently performed interventions were PCI and CABG. In the case of PCI, the majority were performed as emergency procedures, with CABG the split was balanced between emergency and elective procedures. This is in contrast with the other interventions which were primarily elective. The highest cost was seen in emergency bed days.

These results illustrate how committing resources at early stages of the pathway is likely to lead to speedier diagnosis and treatment, securing improved patient outcomes and avoiding the need for more expensive interventions further down the pathway. The aim is to evaluate medium and long-terms benefits, with focus on resource utilisation being a cost analysis rather than cost-effectiveness, which considers differences in costs and differences in patient outcomes (clinical, quality of life, mortality). Patient outcomes are featured alongside the resource differences. The deprivation breakdown revealed that people from more deprived areas had

**Table 9. Percentage of deaths before end of follow-up.**

| Intervention | Death before end of follow-up (%) | |
|---|---|---|
| | **Alive** | **Dead** |
| Electrophysiology [EP] ablations—complex | 99.24 | 0.76 |
| Electrophysiology [EP] ablations—standard | 97.54 | 2.46 |
| Electrophysiology [EP] study | 96.00 | 4.00 |
| Cardiac surgery—coronary artery bypass graft [CABG] | 92.44 | 7.56 |
| Cardiac device implants (pacemaker or defibrillator) | 90.93 | 9.07 |
| Percutaneous coronary intervention [PCI] | 91.62 | 8.38 |
| Transcatheter Aortic Valve Implantation [TAVI] | 78.40 | 21.60 |
| Cardiac surgery—valve replacement | 87.69 | 12.31 |

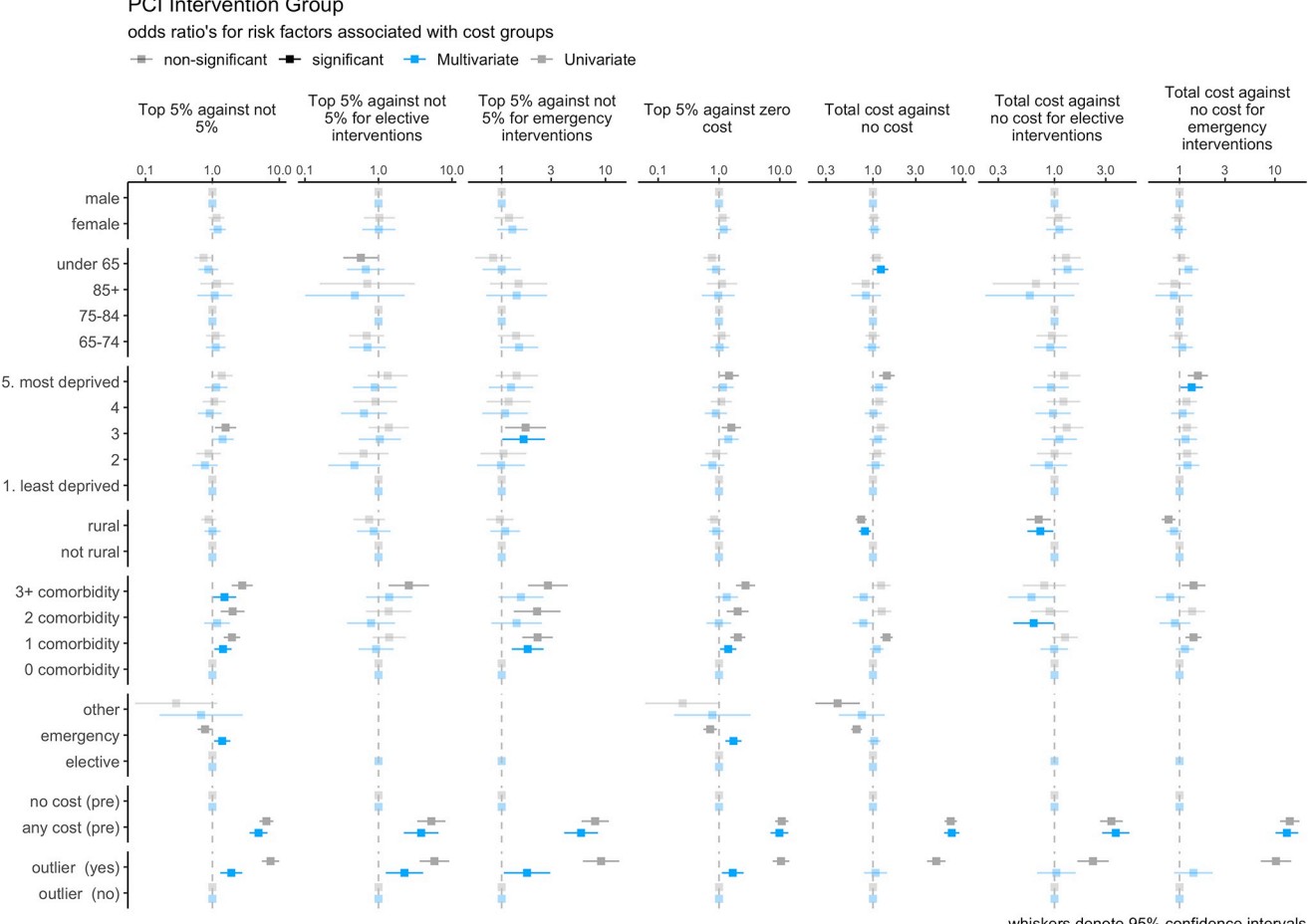

**Fig 9. PCI intervention group—Odds ratios for associated risk factor.**

lower costs before but higher costs after the intervention. Mechanisms which drive associations between deprivation and higher cost health resource use are complex and inter-related. Previous studies have shown that patients with higher levels of deprivation use their GP to a similar level as those living in lesser deprived areas but have higher unplanned care utilisation rates resulting in higher total cost of care per person [29]. Given the type of interventions it is perhaps not wise to assume lower costs will occur post intervention for all patients, but does highlight possible points in the pathway for interventions which may lower costs, such as targeted policies to increase early identification and referral for patients in more deprived communities.

On applying unit cost, hospital bed day costs become amplified due to having a higher relative unit cost. In general, primary care costs were relatively small, but it is worth noting these were derived from events identifiable as physical visits. The greatest costs appear to come from emergency bed days. Other studies have shown marginal cost reduction in healthcare usage following increased expenditure [30], but the picture is nuanced [31], therefore knowing where to target expenditure is valuable knowledge.

When examining specific interventions, in the case of elective EP interventions, there were more elective events after the intervention. The picture is less clear following emergency EP

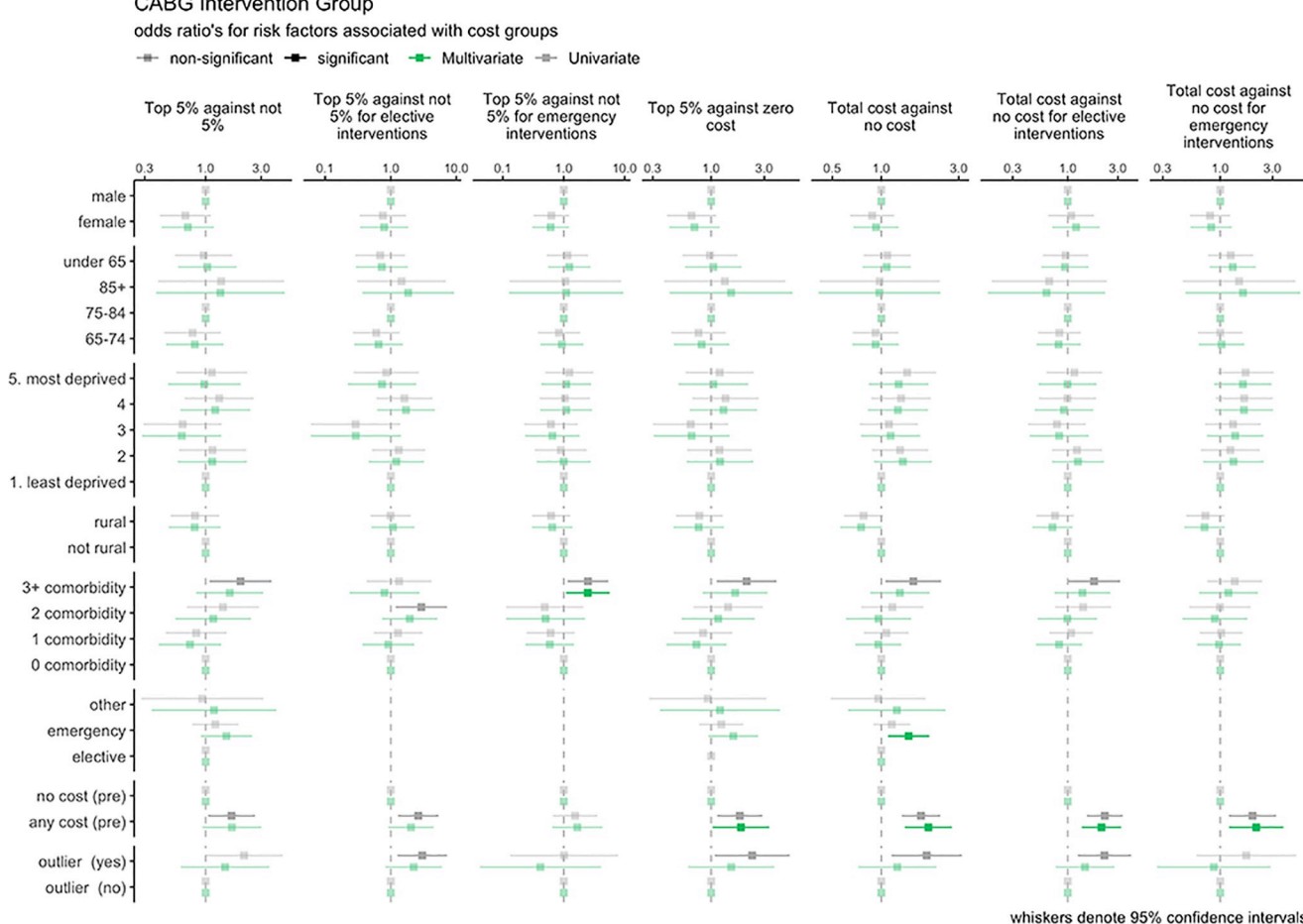

**Fig 10. CABG intervention group—Odds ratios for associated risk factor.**

interventions. CABG, PCI and Valve surgery had broadly similar distributions before and after in both elective and emergency cohorts. Cardiac devices showed a slight trend towards more elective bed days in the elective cohort, whereas the reverse was seen for TAVI.

Our method of evaluating healthcare resource use highlights differences in cost profiles between patients receiving specialised interventions, particularly between those treated following an emergency admission and those treated following an elective admission. These different pathways can be considered proactive or reactive treatment interventions.

Our study has shown that proactive patient management in elective intervention reduces subsequent costs post-intervention, whereby the profile moves towards more representation of elective bed days. Thus providing evidence that may incentivise healthcare providers to identify and treat patients proactively.

The study's strengths include using routinely collected data at the population level and applying the same method across different interventions to facilitate direct comparison. Challenges arise in combining different health outcomes to create an overview of the impact on all services. We followed the data in an unbiased way from first level analysis, noticing the significance of high cost versus zero cost and allowing this to inform a logistic regression analysis comparing the characteristics of patients in these cost groups.

Study limitations include basing costs on averages. At an individual level, there is variation in costs of individual health resource use, but since we are able to look at the whole population, the sum total of the average is representative of actual costs incurred by the health service. Another limitation is the inability to validate coding completeness within the SAIL Databank, with a low number of TAVI procedures for example, which may be accounted for by a lack of accurate coding for that particular intervention. The GP data (WLGP) does not provide 100% coverage, but given the very low number of GP events shown in the results, and the substantially lower cost for GP resource use compared to secondary care, it is unlikely that additional GP data will change the results.

In conclusion, we have shown that early investment in the pathway could potentially reduce later costs. By examining the whole pathway, we can understand the main influences and identify the part of the pathway that would most benefit from investment or change. Allowing the data to lead can help reduce preconceived biases. Deprivation is a key driver in cost variation, and failure to access services in more deprived areas is seen. To understand cost variation, there is a need to better understand inequality and inequity of access to services.

Future research in this area will look at equity of access and the outcomes relating to proactive and reactive management.

## Supporting information

**S1 Table. Significant associations to 5% level from unadjusted univariate analysis.** (DOCX)

**S2 Table. Significant associations to 5% level from multivariate analysis.** (DOCX)

**S3 Table. Codes used to define interventions.** (DOCX)

**S4 Table. Codes used to define conditions.** (DOCX)

## Acknowledgments

The authors would like to thank Ronan Lyons (SAIL Databank), Ceri Phillips (NHS Wales), Karla Williams (WHSSC), and Richard Palmer (WHSSC) for their valuable contribution to the project. This study uses anonymised data held in the Secure Anonymised Information Linkage (SAIL) Databank. We would like to acknowledge all the data providers who make anonymised data available for research.

## Author Contributions

**Conceptualization:** Ashley Akbari, Kendal Smith, Kerryn Lutchman Singh.

**Data curation:** Gareth Davies.

**Formal analysis:** Gareth Davies, Lloyd Evans.

**Funding acquisition:** Kendal Smith, Kerryn Lutchman Singh.

**Methodology:** Rowena Bailey, Kendal Smith, Kerryn Lutchman Singh.

**Project administration:** Kendal Smith, Kerryn Lutchman Singh.

**Supervision:** Ashley Akbari, Rowena Bailey, Kendal Smith, Kerryn Lutchman Singh.

**Visualization:** Rowena Bailey, Lloyd Evans, Kendal Smith.

**Writing – original draft:** Gareth Davies.

**Writing – review & editing:** Ashley Akbari, Rowena Bailey, Lloyd Evans, Kendal Smith, Jonathan Goodfellow, Michael Thomas, Kerryn Lutchman Singh.

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
