## [Decision Letter · Decision Letter 0]

11 Sep 2023

PONE-D-23-25402Cardiac interventions in Wales: A comparison of benefits between NHS Wales specialtiesPLOS ONE

Dear Dr. Davies,

Thank you for submitting your manuscript to PLOS ONE. After careful consideration, we feel that it has merit but does not fully meet PLOS ONE’s publication criteria as it currently stands. Therefore, we invite you to submit a revised version of the manuscript that addresses the points raised during the review process.

We look forward to receiving your revised manuscript.

Kind regards,

Amirmohammad Khalaji

Academic Editor

PLOS ONE

Reviewers' comments:

Reviewer's Responses to Questions

**Comments to the Author**

1. Is the manuscript technically sound, and do the data support the conclusions?

Reviewer #1: Yes

Reviewer #2: Yes

2. Has the statistical analysis been performed appropriately and rigorously? 

Reviewer #1: Yes

Reviewer #2: Yes

3. Have the authors made all data underlying the findings in their manuscript fully available?

Reviewer #1: Yes

Reviewer #2: No

4. Is the manuscript presented in an intelligible fashion and written in standard English?

Reviewer #1: No

Reviewer #2: Yes

5. Review Comments to the Author

Reviewer #1: Davies et al. have performed a study on the assessment of specialized healthcare service interventions in Wales. There are some points that need to be considered.

MAJOR Remarks:

- The English language of the manuscript can be improved; for instance, in lines 33 and 34. Significant changes should be made all over the manuscript.

- The rationale and aim of this study should be clearly described at the end of the introduction.

- The first paragraph of the discussion should highlight the main findings of the study summarized.

- Discussion should focus on the comparison of these findings with similar studies.

MINOR Remarks:

- In the abstract, what do the authors mean by comorbidity and deprivation?

Reviewer #2: In this study, authors have assessed the cost-effectiveness of cardiac interventions and identified the effect of different factors and disparities on the overall treatment cost using comprehensive electronic health record data.

This study addresses the critical issue of equitable access to specialized healthcare services, provides evidence-based insights, and has the potential to influence healthcare policy and resource allocation. It contributes valuable information to the ongoing efforts to improve healthcare delivery and ensure that all population members have fair and equal access to needed medical interventions.

Nevertheless, there are several points that can improve the quality of the manuscript.

1- The introduction could benefit from a more concise statement of the research question or hypothesis the study aims to address at the end of this section.

2- In the methods section, the authors must discuss in more detail how the missing data were addressed (patients with follow-ups or interventions in centers outside the current HER database) and if any sensitivity analyses were performed.

3- The authors need to explain how the covariates were chosen for the adjusted analysis.

4- It is better to perform a survival analysis and present KM curves for survival during the follow-up time. If this is not possible, the limitations section should mention the reason.

5- In the discussion section, the authors need to discuss possible reasons behind the study findings. For example. The reason behind the observed association between deprivation and costs.

6- I suggest that authors compare their results to previous studies in the discussion section (Survival rates, elective/emergency ratios, Costs, ...)

7- The authors should briefly explain how early investment in the pathways can reduce overall costs and, if possible, add some suggestions on improving cost-effectiveness according to the study results.

8- Some sentences in the introduction are quite long and complex, which may hinder readability. Consider breaking down some of these sentences into smaller, more digestible portions for improved clarity.

9- Please proofread the manuscript regarding Punctuation, consistency in Hyphen Usage, and Use of acronyms.

6. PLOS authors have the option to publish the peer review history of their article (what does this mean?). If published, this will include your full peer review and any attached files.

Reviewer #1: No

Reviewer #2: **Yes: **Aryan Ayati

---

## [Author Response · Author response to Decision Letter 0]

26 Oct 2023

Re: PONE-D-23-25402

Cardiac interventions in Wales: A comparison of benefits between NHS Wales specialties

Thank you for your email dated 11th September 2023 inviting us to revise our paper. We are very grateful for the expert feedback, which we have carefully considered and addressed.

Please see "letter of rebuttal" for detailed response to each point raised.

---

## [Decision Letter · Decision Letter 1]

13 Nov 2023

PONE-D-23-25402R1Cardiac interventions in Wales: A comparison of benefits between NHS Wales specialtiesPLOS ONE

Dear Dr. Davies,

Thank you for submitting your manuscript to PLOS ONE. After careful consideration, we feel that it has merit but does not fully meet PLOS ONE’s publication criteria as it currently stands. Therefore, we invite you to submit a revised version of the manuscript that addresses the points raised during the review process.

We look forward to receiving your revised manuscript.

Kind regards,

Amirmohammad Khalaji

Academic Editor

PLOS ONE

Journal Requirements:

Reviewers' comments:

Reviewer's Responses to Questions

**Comments to the Author**

1. If the authors have adequately addressed your comments raised in a previous round of review and you feel that this manuscript is now acceptable for publication, you may indicate that here to bypass the “Comments to the Author” section, enter your conflict of interest statement in the “Confidential to Editor” section, and submit your "Accept" recommendation.

Reviewer #1: All comments have been addressed

Reviewer #2: (No Response)

2. Is the manuscript technically sound, and do the data support the conclusions?

Reviewer #1: Yes

Reviewer #2: Yes

3. Has the statistical analysis been performed appropriately and rigorously? 

Reviewer #1: Yes

Reviewer #2: Yes

4. Have the authors made all data underlying the findings in their manuscript fully available?

Reviewer #1: No

Reviewer #2: No

5. Is the manuscript presented in an intelligible fashion and written in standard English?

Reviewer #1: Yes

Reviewer #2: Yes

6. Review Comments to the Author

Reviewer #1: (No Response)

Reviewer #2: I thank the authors for adequately addressing the comments. I believe the manuscript looks much better now. I only have two remaining comments:

- In response to reviewer comments, the authors have moved almost all of the results section to the beginning of the discussion. I suggest these parts remain in the results section and only a summary and the highlights of these results be added to the beginning of the discussion in one short paragraph.

- Additional fluency modifications are required, especially in lines 162-165.

7. PLOS authors have the option to publish the peer review history of their article (what does this mean?). If published, this will include your full peer review and any attached files.

Reviewer #1: No

Reviewer #2: **Yes: **Aryan Ayati

---

## [Author Response · Author response to Decision Letter 1]

12 Dec 2023

Re: PONE-D-23-25402R1

Cardiac interventions in Wales: A comparison of benefits between NHS Wales specialties

Thank you for inviting us to modify the first revision of our paper. Again we are very grateful for the expert feedback.

We detail the comments needing action, followed by our responses:

Reviewer #2: I thank the authors for adequately addressing the comments. I believe the manuscript looks much better now. I only have two remaining comments:

- In response to reviewer comments, the authors have moved almost all of the results section to the beginning of the discussion. I suggest these parts remain in the results section and only a summary and the highlights of these results be added to the beginning of the discussion in one short paragraph.

The results at the start of the discussion have been moved back into the results section, and a more concise paragraph summarising the main findings has been added to the beginning of the discussion section.

- Additional fluency modifications are required, especially in lines 162-165.

Lines 162-165 have been re-written to improve readability.

Yours Sincerely,

Gareth Davies (Swansea University), on behalf of the co-authors

---

## [Decision Letter · Decision Letter 2]

27 Dec 2023

Cardiac interventions in Wales: A comparison of benefits between NHS Wales specialties

PONE-D-23-25402R2

Dear Dr. Davies,

We’re pleased to inform you that your manuscript has been judged scientifically suitable for publication and will be formally accepted for publication once it meets all outstanding technical requirements.

Kind regards,

Amirmohammad Khalaji

Academic Editor

PLOS ONE

Additional Editor Comments (optional):

Reviewers' comments:

Reviewer's Responses to Questions

**Comments to the Author**

1. If the authors have adequately addressed your comments raised in a previous round of review and you feel that this manuscript is now acceptable for publication, you may indicate that here to bypass the “Comments to the Author” section, enter your conflict of interest statement in the “Confidential to Editor” section, and submit your "Accept" recommendation.

Reviewer #2: All comments have been addressed

2. Is the manuscript technically sound, and do the data support the conclusions?

Reviewer #2: Yes

3. Has the statistical analysis been performed appropriately and rigorously? 

Reviewer #2: Yes

4. Have the authors made all data underlying the findings in their manuscript fully available?

Reviewer #2: No

5. Is the manuscript presented in an intelligible fashion and written in standard English?

Reviewer #2: Yes

6. Review Comments to the Author

Reviewer #2: (No Response)

7. PLOS authors have the option to publish the peer review history of their article (what does this mean?). If published, this will include your full peer review and any attached files.

Reviewer #2: **Yes: **Aryan Ayati

---

## [Editor Report · Acceptance letter]

31 Jan 2024

PONE-D-23-25402R2 

PLOS ONE

Dear Dr. Davies, 

I'm pleased to inform you that your manuscript has been deemed suitable for publication in PLOS ONE. Congratulations! Your manuscript is now being handed over to our production team.

Kind regards, 

on behalf of

Dr. Amirmohammad Khalaji 

Academic Editor

PLOS ONE